



# Heterogeneous grain growth and vertical mass transfer within a snow layer under temperature gradient

Lisa Bouvet[1,2], Neige Calonne[1], Frédéric Flin[1], and Christian Geindreau[2]

[1]Univ. Grenoble Alpes, Université de Toulouse, Météo-France, CNRS, CNRM, Centre d'Études de la Neige, Grenoble, France
[2]Université Grenoble Alpes, Grenoble INP, 3SR, CNRS, Grenoble, France

**Correspondence:** Lisa Bouvet (lisa.bouvet@univ-grenoble-alpes.fr)

**Abstract.**

Inside a snow cover, metamorphism plays a key role in snow evolution at different scales. This study focuses on the impact of temperature gradient metamorphism on a snow layer in its vertical extent. To this end, two cold-laboratory experiments were conducted to monitor a snow layer evolving under $100$ K m$^{-1}$ using X-ray tomography and environmental sensors. The first
experiment shows that snow evolves differently in the vertical: at the end, coarser depth hoar are found in the center part of the layer, with covariance lengths about 50% higher, compared to the top and bottom areas. We show that this heterogeneous grain growth could be related to the temperature profile and the associated crystal growth regimes, and to the profile of vapor supersaturation. In the second experiment, a non-disturbing sampling method was applied to enable a precise observation of the basal mass transfer in the case of dry boundary conditions. An air gap, characterized by a sharp drop in density, developed
at the base and reached more than 3 mm after a month. The two reported phenomena, heterogeneous grain growth and basal mass loss, create heterogeneities in snow - in terms of density, grain and pore size, and ice morphology - from an initial homogeneous layer. Finally, we report the formation of hard depth hoar associated with an increase of SSA observed in the second experiment with higher initial density. These micro-scale effects may strongly impact the snowpack behavior, e.g. for snow transport processes or snow mechanics.

**Introduction**

Temperature gradients are frequently encountered in natural snowpacks with a warmer base and a colder snow surface depending on the atmospheric conditions. Temperature gradient induces disequilibrium in water vapor pressure that forces sublimation of the warmer part of the grains and vapor deposition at their colder extremities. This leads to an overall vertical vapor flux and rapid changes of the snow microstructure, called temperature gradient metamorphism (TGM) (e.g., Yosida, 1955; Pinzer
et al., 2012). TGM transforms snow grains into faceted crystals (FC) in the case of moderate gradients (below $20$ K m$^{-1}$) and depth hoar (DH) for stronger gradients (see *The International Classification for Seasonal Snow on the Ground* by Fierz et al., 2009). Those grain types are characterized by angular shapes and coarser grains, often loosely bonded. They constitute typical weak layers involved in slab avalanches (Schweizer et al., 2003). Besides temperature gradient, other parameters such as the pore size and the temperature can impact the evolution and lead to various sub-types of depth hoar, such as hard depth hoar



(Akitaya, 1974; Marbouty, 1980; Pfeffer and Mrugala, 2002).

TGM has been investigated for several decades. The basic mechanisms of vapor transport and grain growth have been contoured by field observations (e.g., Fukuzawa and Akitaya, 1993; Sturm and Benson, 1997), and by laboratory experiments (e.g., Yosida, 1955; Akitaya, 1974; Marbouty, 1980; Fukuzawa and Akitaya, 1993; Kamata and Sato, 2007; Satyawali et al., 2008) mainly based on photographs and thin sections. More recent studies have widely used the X-ray absorption micro-tomography

technology to get 3D precise visualizations of the snow microstructure (e.g., Brzoska et al., 1999; Kaempfer and Schneebeli, 2007; Chen and Baker, 2010; Ishimoto et al., 2018). For snow evolving under TGM, two different methods were used, the static approach, where a snow layer is temperature-controlled and snow samples are extracted at regular time intervals (e.g., Flin and Brzoska, 2008; Srivastava et al., 2010; Calonne et al., 2014), and the in-vivo approach, where the same snow sample is monitored by time-lapse tomography using an environmental cell (e.g., Pinzer and Schneebeli, 2009; Pinzer et al., 2012;

Calonne et al., 2015a; Hammonds et al., 2015; Wiese, 2017; Granger et al., 2021; Li and Baker, 2022). These approaches enable a better understanding of TGM time evolution and highlight the role of snow microstructure on its physical and mechanical properties. However, most studies focus on small snow samples (about 1 cm$^3$), often located in the center of the snow layer. The microstructural evolution of such samples is commonly considered as representative of the evolution of the snow layer. However, conditions can vary substantially inside a snow layer especially along the vertical, such as the temperature and

the vapor pressure, depending on the boundary conditions. It seems thus important to consider the evolution within the whole vertical extent. We investigate here two phenomena induced by temperature gradient along the vertical direction: the basal mass transfer and the influence of temperature and vapor transport on the microstructure evolution.

One process attributed to strong TGM is the loss of ice mass close to the soil interface (Kamata and Sato, 2007; Wiese, 2017; Domine et al., 2019). In natural conditions, the warm area is located at the ground interface and the vertical vapor flux can dry

the basal area and transport the ice to a higher area of the snowpack, leading to a basal low-density layer. This phenomenon was first reported in the field in sub-arctic and arctic snowpack, which undergoes high temperature gradients. Sturm and Benson (1997) observed a poorly-bonded weak layer near the ground interface from traditional snow profiles. More recently, Domine et al. (2019) reported a low-density layer of the order of 10 cm height at the bottom of the arctic snowpack of total height of 40 cm. To our knowledge, only two experimental studies were dedicated to the quantification of this basal mass loss. Kamata and

Sato (2007) monitored the evolution of a 10 cm height snow layer subjected to an extreme temperature gradient of 530 K m$^{-1}$ during 5.5 days. Density values were obtained by weighing sub-layers extracted from the total layer with a vertical resolution of 2.5 cm. The density of the first 2.5 cm (bottom) decreases during the experiment whereas it increases in the other sub-layers. Later, Wiese (2017) captured the drying at the snow-soil interface more precisely by in-vivo tomography and investigated the influence of the lower boundary conditions. In the presence of an artificial humid soil (iced glass beads) and a temperature

gradient of 100 K m$^{-1}$, they observed the formation of a low density layer as well as the partial drying of the soil, for small samples of 2 cm height and 0.5 cm diameter. As pointed out by Domine et al. (2019), this decrease in density caused by vapor transport, is not captured by the snow cover models, such as Crocus (Vionnet et al., 2012) or Snowpack (Lehning et al., 2002). It could however have an impact on the way several snow properties are today represented in those models, which highlights the need to better characterize and quantify this process.



Different factors impact the evolution of snow microstructure during TGM, besides the temperature gradient itself (Akitaya, 1974; Marbouty, 1980; Pfeffer and Mrugala, 2002). At higher temperatures, the rate of metamorphism is enhanced, as the deposition-sublimation processes are more active, and inversely. This dependency was well illustrated by Kamata and Sato (2007), who presented pictures of the final and initial microstructures for several heights of the snow layer undergoing a TGM of 530 K m$^{-1}$. The snow evolving at about -55°C (upper part of the layer) remained at its initial stage and barely evolved, whereas snow evolving at -15°C (lower part) transformed in depth hoar of about 1 mm in size. Temperature range and water vapor also influence the shape of the growing crystals. This dependency has been widely studied in the case of fresh snow crystals growing in the atmosphere, as described by the Nakaya diagram (Nakaya, 1954; Hallett and Mason, 1958; Kobayashi, 1961; Rottner and Vali, 1974; Yokoyama and Kuroda, 1990; Furukawa and Wettlaufer, 2007; Bailey and Hallett, 2009; Libbrecht, 2021), but little for snow grains inside a snow cover. Akitaya (1974) observed similar growth habits for depth hoar growing inside a snow cover with shapes like plates, needles, or cups depending on temperatures and temperature gradient degree using photographs. More recently Ozeki et al. (2020) mentioned similar patterns for surface hoar. On the field, Sturm and Benson (1997) observed the grains morphology from snow profiles of subarctic snowpack and reported rather column-like of planar-like depth hoar at different heights of the snow layer, which also may imply an effect of snow temperature and saturation conditions. Another impacting factor is the initial density and pore size, so that small pore space or high density impede the growth of large depth hoar crystals as they are more constrained in space (Akitaya, 1974; Pfeffer and Mrugala, 2002).

The present work aims at observing the mass transfer and microstructure evolution of a snow layer under TGM along the vertical. Those evolution are investigated with two TGM experiments at 100 K m$^{-1}$ during 20 and 28 days, based on X-ray tomography. Temperature and humidity profiles were recorded during the first experiment to monitor the environmental conditions of the snow layer. To improve the reliability of the tomographic acquisitions at the base of the layer, a new non-disturbing sampling method was developed in the second experiment, thus capturing precisely the mass transfer at the snow base. Vertical profiles of microstructural properties, such as density, specific surface area (SSA), covariance length, and structural anisotropy were computed from the 3D tomographic images of each experiment. The vertical evolution of the snow microstructure is analyzed based on the microstructural properties and the segmented images. The basal mass loss is quantified through the density profile evolution and physical processes to explain these features are explored.

The article is organized as follows. The experimental setup, the image processing, and the analysis methods are presented in Section 1. The results of the environmental sensors and of the micro-tomographies are shown in Section 2. In Section 3, the evolution of the grain morphology, the vertical mass transfer, and the hard depth hoar formation are discussed for both experiments. Finally, Section 4 concludes the manuscript.

# 1 Method

## 1.1 Experimental setup

We carried out two experiments that follow the static method to monitor the evolution of a snow layer artificially created in a temperature gradient box. On both experiments, snow profiles are scanned at regular intervals with a DeskTom130 tomograph



**Table 1.** Overview of the experimental settings.

| Experiment | A (fully instrumented experiment) | B (non-disturbing sampling experiment) |
|---|---|---|
| initial density (kg m$^{-3}$) | 210 | 287 |
| initial snow type | DF | RG |
| grain size (mm) | $\sim 0.3$ | $\sim 0.3$ |
| snow layer height (cm) | 13.5 | 7.7 |
| temperature gradient (K m$^{-1}$) | 93 | 103 |
| snow base temperature (°C) | -3.1 | -6.5 |
| snow surface temperature (°C) | -15.6 | -14.5 |
| measurements | 9 tomographies at 21 µm ($2.1 \times 2.1 \times 12$ cm$^3$) 17 tomographies at 8 µm ($1 \times 1 \times 1$ cm$^3$) Temperature and humidity sensors (SHT15, SHT25, PT100) | 4 tomographies at 10 µm ($1.3 \times 1.3 \times 4.2$ cm$^3$) |
| duration (days) | 20 | 28 |

(RX Solutions) placed inside a cold room at -10°C. In Experiment A, the layer is thicker, leading to a better appreciation of the temperature influence on metamorphism. Temperature and humidity profiles were thus recorded. As in this experiment, the sampling method damaged the lower millimeters, the method was improved in Experiment B: cylindrical samplers, closed at the bottom, were placed in the snow layer at the initial stage and horizontally dug out when needed. An overview of the experimental settings used for both experiments is provided in Table 1.

### 1.1.1 Experiment A: fully instrumented experiment

Natural fresh snow was collected at St Pancrasse (1000 m, French Alps) in April 2021 and stored at -20°C for 3 weeks. This snow was then sieved using a 1.6 mm diameter sieve in a cold room at -10°C to obtain a horizontal snow slab of 97 cm length, 58 cm width, and 14.5 cm height, composed of decomposing and fragmented precipitation particles (DF) at 210 kg m$^{-3}$. The snow layer was confined at the base and the top between two copper plates whose temperature was controlled by a thermo-regulated fluid circulation. The whole system was insulated with 8 cm thick polystyrene plates. A photograph of the experimental setup and a schematic representation of the top and side view is given in Fig. 1. Isothermal conditions at -10°C were applied to the snow slab for 21 hours. This aimed at sintering snow grains whose bonds have been destroyed by sieving. After the isothermal stage, the upper copper plate was lowered by a cm to close the gap caused by the snow settling. During the following 20 days, the temperature of the cold room was held at -10°C and the upper and lower copper plates were maintained at -15.6 and -3.1°C, respectively, generating a steady vertical temperature gradient of about 93 K m$^{-1}$ through the snow layer.





During the sieving process, three types of temperature and humidity sensors were placed in the snow layer at regular heights
to obtain three vertical profiles:

- a profile of 7 PT100, resistance thermometers made of platinum. Their temperature range varies from -50°C to 200°C,
  they have a 3 mm diameter and are 25 mm long, and their accuracy is ± 0.2°C.

- a profile of 5 SHT15 (Sensirion), sensors that contains a capacitive type humidity sensor and a band gap temperature
  sensor. Their temperature operating range is -40°C to 128°C, their dimensions are 32 x 17 x 1 mm, and their accuracy is
  ± 4.5% relative humidity (RH) and ± 0.5°C.

- a profile of 5 SHT25 (Sensirion), sensors that also contains a capacitive type humidity sensor and a band gap temperature
  sensor. Their temperature operating range is -40°C to 125°C, their dimensions are 3 x 3 x 1 mm, and their accuracy is ±
  1.8% RH and ± 0.2°C.

The sensors are marketed with a humidity and temperature calibration at ambient conditions ($\sim$ 20°C and $\sim$ 50% RH) and
present large offsets when placed in cold and humid conditions, up to 2°C and 7% RH. A calibration was conducted between
-4°C and -14°C, and between 85% and 100% RH. In terms of temperature, an offset was applied to each sensor between -0.1
and 1.2°C. For the humidity, as the error is correlated to the temperature, a linear correction was applied by type of sensor
(SHT25 and SHT15), in our range of temperature, the correction was between 0 and 8%. The capacitive humidity sensors
provide relative humidity above liquid water in %, which is defined as the ratio of the vapor density of the air to the water
saturation vapor density. To derive water vapor density in kg m$^{-3}$, the following equation was used to obtain the relative
humidity above ice:

$$\text{RH}_{\text{ice}} = \text{RH}_{\text{water}} \times e^{17.62 \times T/(243.12+T)}/e^{22.46 \times T/(272.62+T)} \tag{1}$$

as well as the Clausius-Clapeyron equation (linking the saturation water vapor density to the temperature) to derive the water
vapor density from the relative humidity:

$$\rho_v = \frac{\text{RH}_{\text{ice}}}{100} \times \rho_v^{\text{ref}} \times \exp\left(\left(\frac{1}{T_{\text{ref}}} - \frac{1}{T}\right) \times L_{sg} m/\rho_i k\right) \tag{2}$$

with $\rho_v^{\text{ref}} = 2.173\ 10^{-3}$ kg m$^{-3}$ and $T_{\text{ref}} = 263$ K the reference values, $L_{sg} = 2.6\ 10^9$ J m$^{-3}$ the latent heat of sublimation
of ice, $m = 2.99\ 10^{-26}$ kg the mass of a water molecule, $\rho_i = 917$ kg m$^{-3}$ the ice density and $k = 1.38\ 10^{-23}$ J K$^{-1}$ the
Boltzmann constant.

During the 20 days of the experiment, 9 tomography sessions were done at regular time intervals. It consisted in the sampling
and scanning of one large sample of the whole vertical dimension and two small samples localized in the top and in the bottom
of the snow layer (Fig. 1), leading to a total of 26 samples (only one small sample was collected during the first sampling). For
each sampling, the polystyrene plates and the upper copper plate were temporarily removed in order to access the snow slab,
disturbing momentarily the upper temperature boundary and breaking the ice bounds between the plate and the snow layer. The
samples were vertically extruded at a minimum distance of 7 cm from edges and from regions already sampled. Large samples





were extracted using a PMMA serrated cylindrical core drill with a diameter of 4.5 cm and a height of 14 cm. Small samples
were extracted using a copper cylindrical core drill with a diameter of 1 cm and a length of 2.5 cm. A flat trowel was inserted
in contact to the lower copper plate, the cylinder was pushed in the snow layer from the top, and then carefully extracted and
placed in the tomograph. The very bottom area of the snow was disturbed because of the samplers pressed down in the snow
layer, which crushed the grains against the copper plate, and because of the insertion of the flat trowel at the base of the snow
layer that destroyed 2 to 3 mm of snow. Holes in the snow layer created by the sampling were systematically refilled with snow
to ensure stable temperature and humidity fields. Large samples ($2.1 \times 2.1 \times 12$ cm$^3$) were scanned with a pixel size of 21 µm.
The X-ray tube was powered by a voltage of 60 kV and a current of 260 µA. Small samples ($1 \times 1 \times 1$ cm$^3$) were scanned
with a pixel size of 8 µm, a voltage of 62 kV and a current of 129 µA.

### 1.1.2  Experiment B: non-disturbing sampling experiment

Natural snow was collected at Col de Porte (1325 m, French Alps) in February 2022 and stored at -10°C for 3 days. The snow
was then sieved using a 1.6 mm diameter sieve in a cold room at -10°C to obtain a horizontal snow layer of 97 cm length, 58
cm width, and 7.7 cm height, composed of rounded grains (RG) at 287 kg m$^{-3}$. The same temperature gradient box described
in Experiment A was used. During the following 28 days, the temperature of the cold room was held at -8°C. Temperatures of
the top and lower copper plates were maintained at -15.6 and -3.1°C, respectively.

For this experiment, the sampling method was improved compared to Experiment A, for which the insertion of the cylinder
from the top can disturb the snow, especially at the base of the layer where grains could crush against the copper plate, which
prevents analyzing snow at the base. Here, the new method allows to scan undisturbed snow samples from top to bottom. For
that, before snow sieving, 8 cylinders made of PMMA with a diameter of 4.5 cm and a height of 4.2 cm were placed on the
lower copper plate; the base of the cylinders were closed by a 5 mm thick layer of polystyrene. Cylinders were then filled with
snow and buried in the layer during the sieving process. An illustration and a detailed scheme of the experimental setup is given
in Fig. 2.a and 2.b. The modifications of the temperature field caused by the cylinder and the polystyrene layer were modeled
solving heat transfer equations using the finite element software COMSOL Multiphysics® (Fig. 2.c). This figure shows that
the initial setup on the left, with the polystyrene only inside the cylinders, tends to modify the temperature field. When the
whole copper plate is covered with the polystyrene layer, this effect is reduced, as shown in the right simulation. With this
layer of polystyrene insulating the bottom of the snow, the upper and lower copper plates were maintained at -14.5 and -3.5°C,
respectively, which led to a temperature of -6.5°C at the snow base above the polystyrene layer, leading to a temperature
gradient of about 103 K m$^{-1}$.

Sampling for tomography were performed after 1, 7, 17, and 28 days of temperature gradient. Each time, two samples were
extracted and scanned using X-ray tomography. For sampling, the cylinders, buried in the snow layer, were excavated by
digging horizontally from the front of the snow layer and gently dragged out. The hole was then filled with snow. This method
enables sampling without moving the copper plates and only the front polystyrene plate needs to be temporarily removed. For
tomography, the X-ray tube was powered by a voltage of 70 kV and a current of 114 µA. As both samples of each session were





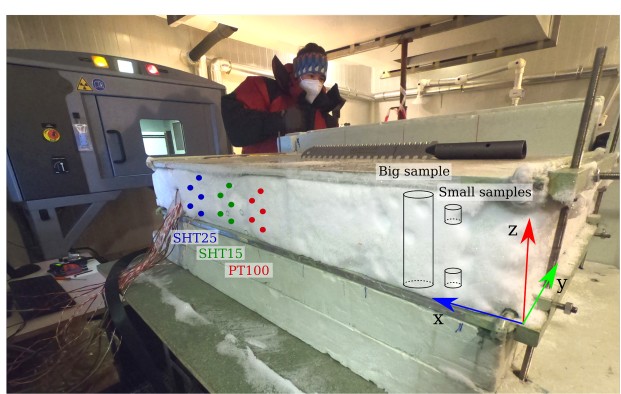

a) Photograph of the experimental setup

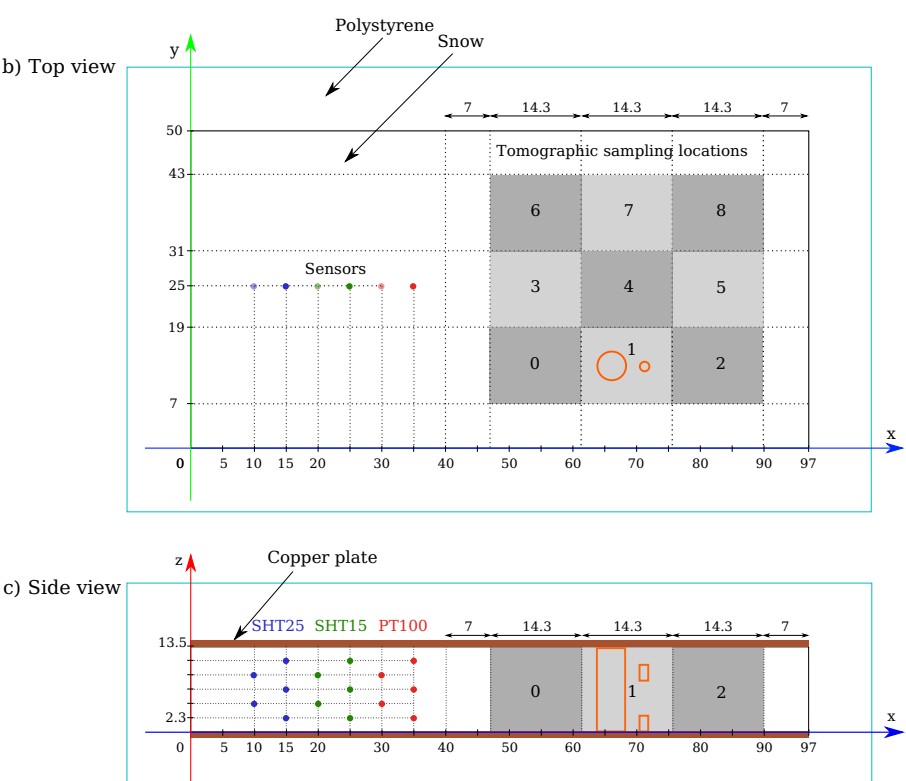

**Figure 1.** (a) Photograph of the experimental setup of Experiment A. (b) Schematic view from above the instrumented box. (c) Schematic view from the side. The locations (in cm) of the 3 types of sensors are shown with blue, green and red dots. The tomography sample locations are numbered from 0 to 8 following the chronology of the samplings. The orange contours represent the tomographies areas for the session 1.



visually consistent, only one scan of each day has been analyzed. A total of 4 scanned samples with a size of $1.3 \times 1.3 \times 4.2$ cm$^3$ and a resolution of 10 µm are presented.

## 1.2 3D image processing and analysis

For both experiments, the tomographies were reconstructed into three-dimensional grey-scale images representing the attenuation coefficients of the different materials composing the samples. The ice phase was segmented from the air phase using the energy-based segmentation algorithm developed by Hagenmuller et al. (2013).

Using the segmented 3D images, properties were computed to characterize the microstructural evolution. The snow density $\rho_s$ (kg m$^{-3}$) was computed using a simple voxel counting algorithm. The specific surface area SSA (m$^2$ kg$^{-1}$), defined as the total surface area of ice per unit of mass was computed using the voxel projection approach (Flin et al., 2011; Dumont et al., 2021). The covariance (or correlation) length $l_c$, which corresponds to the characteristic size of a heterogeneity made of an ice grain and a pore, was calculated along the x-, y- and z- directions of the images, as in the work of Calonne et al. (2014). The structural anisotropy coefficient $\mathcal{A}(l_c)$ was computed from the covariance lengths as $\mathcal{A}(l_c) = 2l_c^z/(l_c^x + l_c^y)$. The microstructure is considered isotropic when $\mathcal{A}(l_c)$ is close to 1 and anisotropic otherwise. $\mathcal{A}(l_c)$ values larger than 1 describe a microstructure that is elongated in the vertical direction and values lower than 1 describe a horizontally-elongated microstructure.

For the small samples of Experiment A, properties were computed on the representative elementary volume (REV) of size $1 \times 1 \times 1$ cm$^3$. For large samples, to observe precisely the vertical mass transfer, the density was computed on each horizontal slice with a running average of 100 voxels applied to the profiles. Volume of computation was thus $2.5 \times 2.5 \times 12$ cm$^3$ for Experiment A and $1.3 \times 1.3 \times 4.2$ cm$^3$ for Experiment B. For SSA, covariance length and anisotropy parameters, those large samples have been divided into subsamples with the size of the REV of the properties, here $2.5 \times 2.5 \times 0.96$ cm$^3$ for Experiment A and $1.3 \times 1.3 \times 0.84$ cm$^3$ for Experiment B.

## 2 Results

### 2.1 Experiment A

#### 2.1.1 Temperature and humidity sensors

The temperature and humidity sensors are used here as control variables to monitor the environmental variables of the snow layer. Figure 3 shows the vertical profile of both parameters, which correspond to the average temperature and humidity of the sensors from day 15 to day 17 of the experiment (longest period in steady state conditions, with no sampling). It is worth to notice that after starting the temperature gradient, temperature and humidity reached a stable state after approximately 5 hours and remained rather constant for the whole duration of the experiment (variations maximum of 1% RH and 0.75°C). However, each time the top copper plate was removed to sample the snow layer, the temperature and humidity fields were slightly perturbed during several hours.

The vertical profile of temperature shows a slight upward convex curve, also observed in early experiments (Yosida, 1955;





a) Photographs of the experimental setup

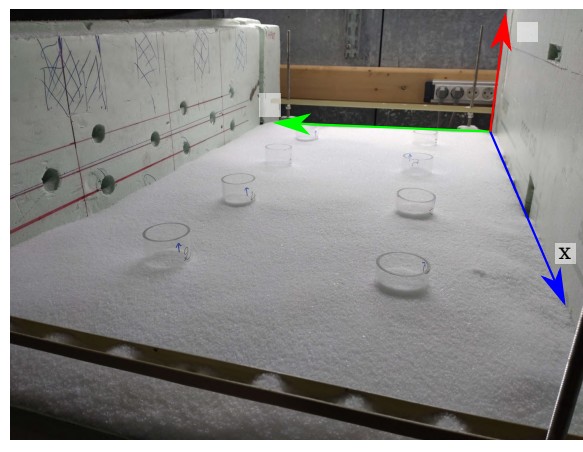
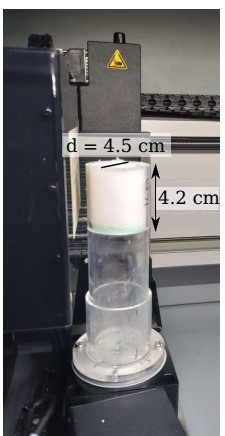

b) Side view

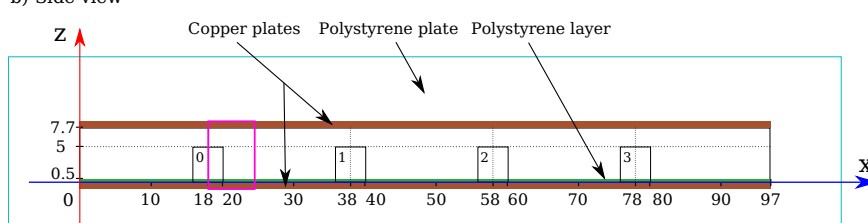

c) Simulation of the temperature field

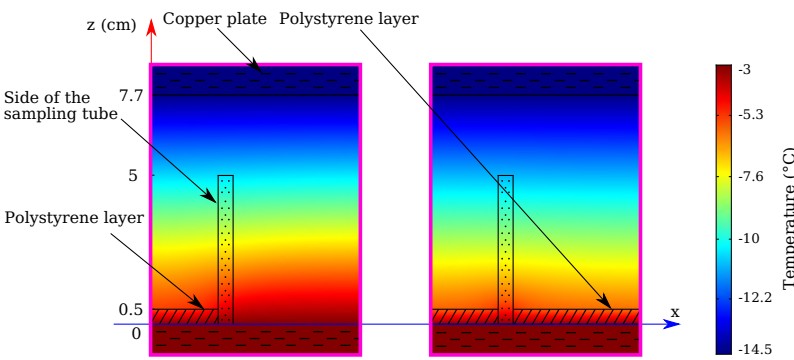

**Figure 2.** (a) Photograph of the experimental setup of Experiment B. (b) Schematic view from the side of the instrumented box. (c) Simulation of the temperature field of the cylinders buried in snow with (left) the polystyrene layer placed only in the cylinders and (right) the polystyrene layer laid on the whole copper plate. The simulated geometry is represented by a pink box in (b) and (c). The simulation was computed with the finite element model COMSOL with axi-symmetric conditions.

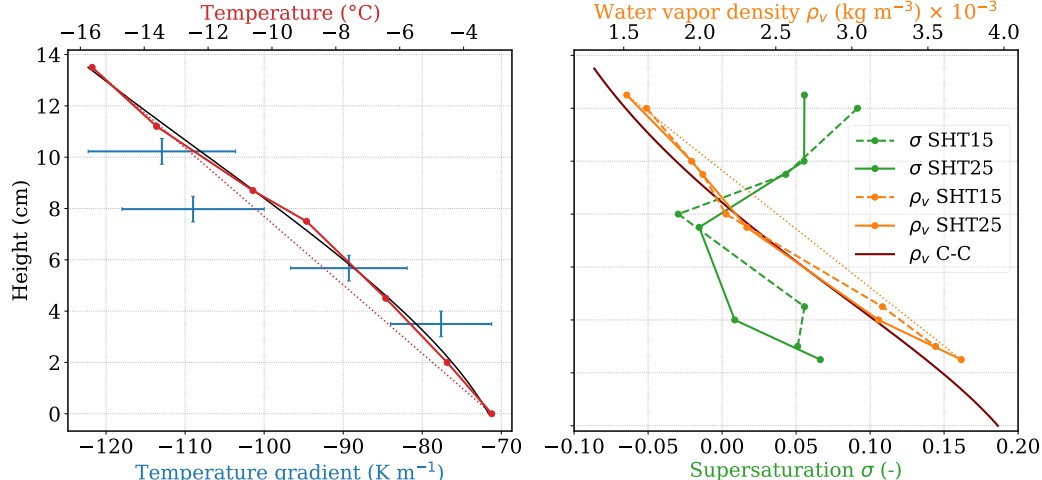

**Figure 3.** Temperature, temperature gradient, water vapor density and supersaturation vertical distributions during the steady condition stage of Experiment A. The black line represents the polynomial fit on the PT100 data. The brown line displays the Clausius-Clapeyron equation applied to the temperature fit. The straight dotted red and orange lines, which connect the higher and lower points of the temperature and the water vapor density, respectively, are presented to help highlighting the shape of the curves.

Fukuzawa and Akitaya, 1993; Sturm and Benson, 1997; Kamata and Sato, 2007), and consistent with the prediction of heat

and vapor transport models, such as the homogenized model of Calonne et al. (2015b). This upward convex curve is due to the latent heat flux from phase change (sublimation and deposition). A mean average on measurement 4 points has been applied to the temperature profile to derive 4 points of measurements for the local temperature gradients. As a result of the curved distribution of temperature, the local temperature gradient is larger on the upper part of the snow layer ($\sim 115$ K m$^{-1}$) than in the bottom part ($\sim 80$ K m$^{-1}$). A similar and more pronounced result is observed in Kamata and Sato (2007) with local

gradients ranging from 1200 K m$^{-1}$ in the cold part to 300 K m$^{-1}$ in the warm part, for a macroscopic gradient of 530 K m$^{-1}$. The vertical profile of water vapor density shows an upward concave curve. Although the sensors present quite a large uncertainty, which can be seen with the differences between the SHT15 and SHT25 profiles, the matching trends suggest a reliable result. This non-linearity is due to the non-linear dependence between the temperature and the saturation vapor density, as predicted by the Clausius-Clapeyron equation, as well as the location of the main vapor sources and sinks. In the figure, the

Clausius-Clapeyron equation applied to the polynomial fit of the PT100 profile (in black in the temperature plot) is shown in brown. It shows a two-point inflection curve, which is the consequence of the upward convex temperature profile (Sturm and Benson, 1997). The humidity profiles deviate from the Clausius-Clapeyron curve, especially far from the center part. The difference between the water vapor density ($\rho_v$) and the saturation water vapor density ($\rho_v^{\mathrm{sat}}(T)$) is described by the supersaturation: $\sigma = (\rho_v - \rho_v^{\mathrm{sat}}(T))/\rho_v^{\mathrm{sat}}(T)$. It presents high values in the top and bottom areas, and values close to 0 in the center

part. The disequilibrium observed at the top and the bottom of the profile might be due to the boundary conditions imposed





by the copper plates. The supersaturation distribution will be used in section 3 to suggest an explanation of the different microstructural features induced by the temperature gradient conditions.

### 2.1.2 Qualitative analysis of the tomographs

Figure 4 presents, in a qualitative way, the microstructure evolution of snow during the 20 days of temperature gradient. Vertical
slices of the snow column of the large samples at 21 μm and horizontal slices of the small samples at 8 μm taken at 0, 7 and 20 days are shown, as well as the associated photographs of grains. Microstructure gradually transforms from decomposing and fragmented precipitation particles (DF) towards depth hoar (DH) and grains and pores are getting larger. Photographs of grains taken with a microscope show the detailed features of the grains, such as the rounded dendrites at day 0, the facets and angular shapes at day 7 and the large hollow striated cup shapes arranged in vertical chains after 20 days. The type of depth
hoar obtained at the end of the experiment is standard depth hoar (formerly called skeleton-type depth hoar; Akitaya, 1974; Fierz et al., 2009).

### 2.1.3 Microstructural properties

Figure 5 presents the vertical profiles of microstructural properties. The small samples are presented in the figure with small vertical bars. The grey area represents the unreliable area at the base of the snow layer caused by the sampling method (see
Sec. 1.1.1).

First, the effect of resolution on the property computations, using both the large samples at 21 μm and the small samples at 8 μm can be observed. Precise comparison are however challenged by the superimposing influence of the spatial variability, which creates variations between samples collected at the same time, but at locations slightly different in the layer. Overall, the small and large samples are of the same order of magnitude, with average differences of 5% for the density, 15% for the SSA,
6% for the covariance length and 2% for the anisotropy parameter.

Looking at the evolution of the properties in time, the density evolution suggests the settling of the snowpack in the first days of the experiment with a slight increase of the density and the loss on about 1 cm height. This rapid settling is mainly due to the isothermal phase before the temperature gradient. After the first 2 days, the density remains stable to the end of the experiment. The SSA decreases overall with time, with mean vertical values evolving from 44.5 m$^2$ kg$^{-1}$ to 17.5 m$^2$ kg$^{-1}$ after 20 days.
Consistently, the mean covariance length increases gradually with time so that the characteristic size of heterogeneities is tripled, from 0.07 mm to 0.23 mm (mean vertical value). The anisotropy coefficient $\mathcal{A}(l_c)$ increases in time, to reach values around 1.2 after 20 days, reflecting the typical elongation of the ice structure along the vertical temperature gradient direction (grain chains).

Then, the evolution of the properties along the vertical dimension is analyzed. First, the density drop observed in the upper
area corresponds to the non-planar transition between the snow and the air. Overall density remains constant along the vertical, around 220 kg m$^{-3}$. Especially, no de-densification is observed at the bottom (taking into account that the snow base could not be imaged). It gets more interesting when looking at SSA and covariance length profiles. Lower SSA, down to 14.5 m$^2$ kg$^{-1}$, and greater covariance length, up to 0.29 mm, are found in the middle part of the snow layer and stand out compared to





**Figure 4.** (a) Vertical slices of the big samples at 21 μm after 0 day, 7 days and 20 days of Experiment A. (b) Horizontal slices of the small samples at 8 μm. The vertical location of the small samples presented here are respectively 8.3 cm, 1.2 cm and 1.2 cm for the samples at 0, 7 and 20 days. Each scale bar represents 5 mm. (c) Photographs of individual grains from the middle of the layer seen by microscope. All samples names and associated parameters are listed in Table 2.



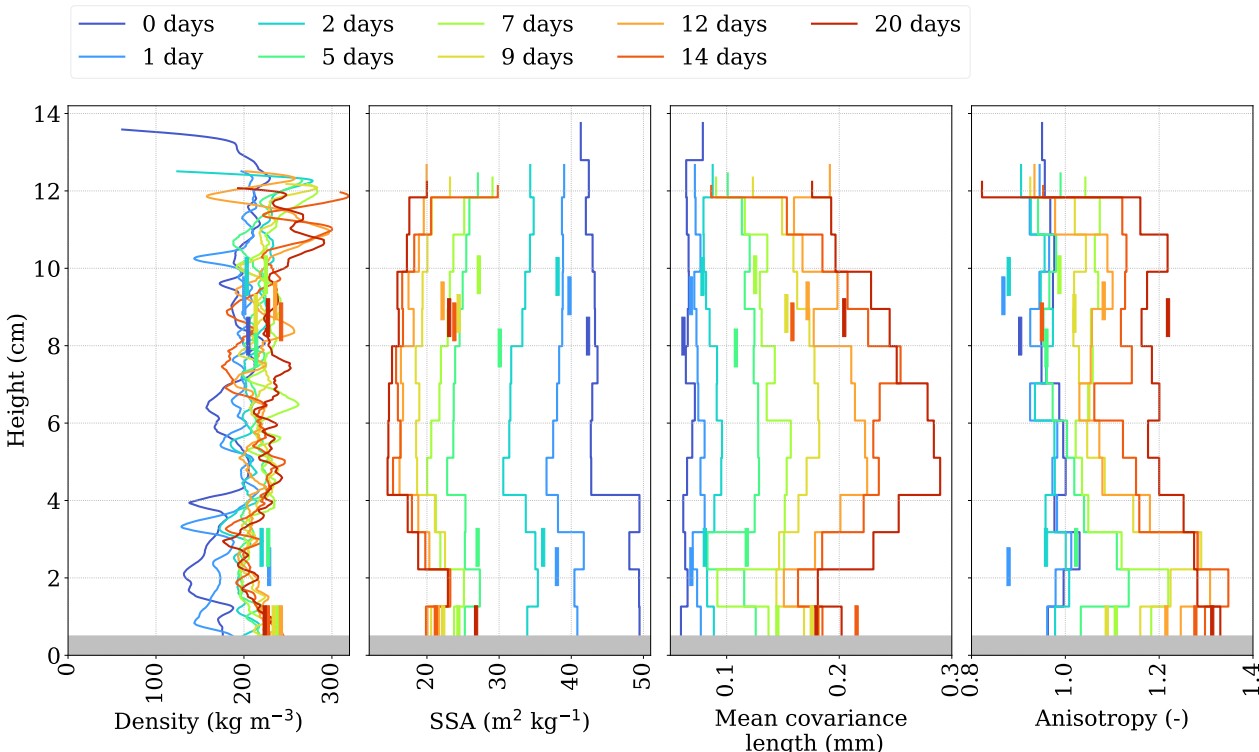

**Figure 5.** Vertical profiles of the microstructural properties of Experiment A. Properties computed from the large samples at 21 μm are shown by step curves and those from the small samples at 8 μm by vertical bars. The density of the samples at 21 μm is computed on each slice with a running average (see Sec. 1.2). The grey area at the bottom represents the disturbed and unreliable area caused by the snow sampling.

values in the top and the bottom part. As these vertical variations are not present at the initial stage, it indicates that the snow microstructure transforms in a different way within the snow layer, more precisely that the size of the ice and air heterogeneities develop a non-monotonous vertical distribution during the experiment. To analyze these differences further, Fig. 6 shows the final state with horizontal slices and 3D views of the snow microstructure in the top, middle and bottom part of the layer, together with the covariance length profile. At the top, the ice grains present many small grains and pores compared to the middle, where larger sparse grains dominate. At the bottom, small grains that have more elongated and dendritic shapes are found, which suggests complex entangled dendritic crystals. This outcome will be further discussed in Section 3.1.

## 2.2 Experiment B

### 2.2.1 Qualitative analysis of the tomographs

A visualization of the microstructure evolution is shown in Fig. 7 with vertical slices of the segmented images, and in Fig. 8 with 3D views of centered sub-samples. As the snow evolves, one can clearly see the transformation from the first step (after 1

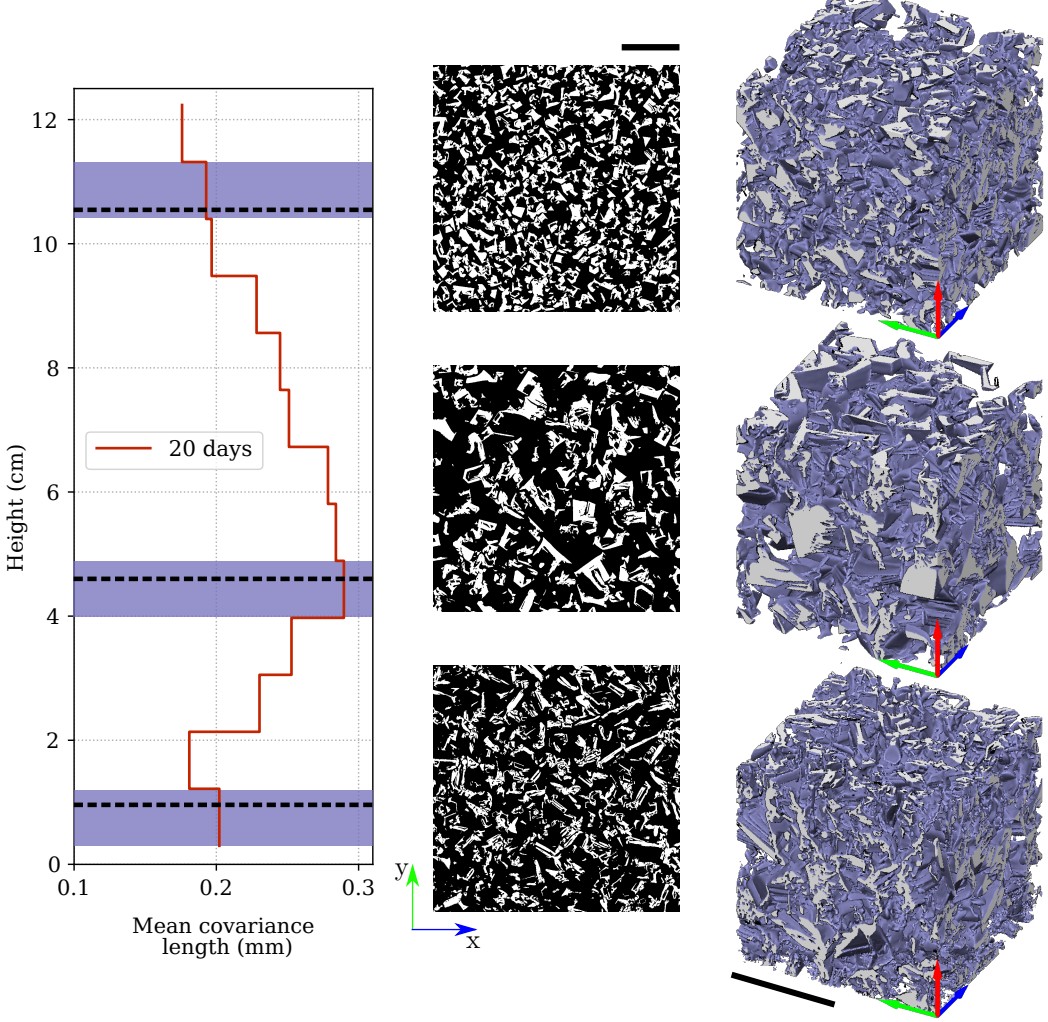

**Figure 6.** Mean covariance length profile, horizontal slices and 3D sub-samples of the large sample at 20 days of Experiment A. Each 3D sub-sample is $0.95 \times 0.95 \times 0.95$ cm$^3$ and is horizontally centered. The vertical locations of the horizontal slices and of the 3D sub-samples are shown respectively with dotted black lines and plain violet rectangles on the covariance length profile. The scale bars represent 5 mm.





**Figure 7.** Vertical slices of the 4 samples of Experiment B. Each scale bar represents 5 mm. All sample names and associated parameters are listed in Table 2.

day of temperature gradient) presenting dense faceted grains (FC) to depth hoar (DH) at 7 days, and finally to the last stages at 17 and 28 days where vertical structures appear and dendritic features develop. The last stage can be characterized as hard depth hoar, which has been previously described as a dense and cohesive structure, with hoar crystals connected by necks and vertical grain-to-grain chains (Akitaya, 1974; Pfeffer and Mrugala, 2002). A hand hardness test was conducted and showed hardness classified as "pencil", corresponding to a high resistance to penetration (Fierz et al., 2009). The figure also provides

a clear insight of the local mass loss at the bottom of the samples, as an air gap gradually forms. Here, unlike in Experiment A, the base of the snow column could be scanned and provides an accurate observation of the bottom interface, as seen on the images.



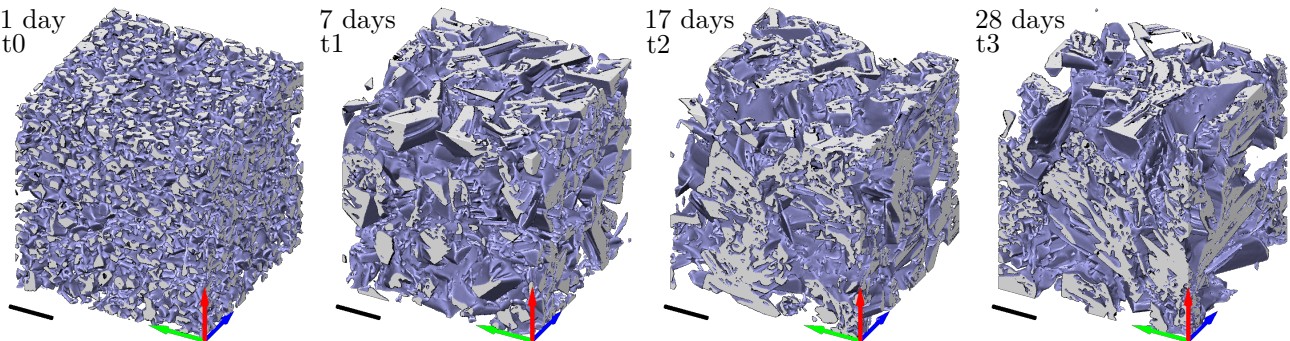

**Figure 8.** 3D representation of sub-samples of experiment B. Each sub-sample is $0.44 \times 0.44 \times 0.44$ cm$^3$, and shows the central part of the imaged snow column. Each scale bar represents 1 mm.

### 2.2.2 Microstructural properties

As for Experiment A, Fig. 9 presents the evolution of the vertical profile of the microstructural properties with time. In terms
of overall temporal evolution, the SSA shows a large decrease for the first days of the experiment, dropping from 39.6 to 27.5 m$^2$ kg$^{-1}$ after 7 days, followed by an increase during the remaining time, up to 33.3 m$^2$ kg$^{-1}$ at the end of the experiment. This increase in SSA can be seen as a specificity of hard depth hoar and will be further discussed in Section 3. The evolution of the covariance length takes place mainly during the first week, where value increase from about 0.07 mm to 0.14 mm in 7 days. Similar correlation lengths, around 0.15 - 0.16 mm, are then found until the end. This evolution of the size of the
heterogeneities can also be observed in Fig. 7 and 8, with a drastic change of the microstructure in the first step, while the microstructure between 17 and 28 days does not show such a pronounced change. Similarly, the anisotropy $\mathcal{A}(l_c)$ increases strongly in the first time step and seems to reach a stable range after 7 days. The mean vertical density considering the full height, decreases from 287 to 273 kg m$^{-3}$, indicating a regular mass loss of 5% for each consecutive time step.

In terms of vertical variations, density shows the most significant evolution. A sharp drop in density localized in a thin layer
at the bottom of the samples is observed. This air gap between the green polystyrene layer and the snow can be seen from the pictures of the snow samples presented in Fig. 10, and by the growing black area at the bottom of the segmented images in Fig. 7. We define the top boundary of the air gap as the height where snow reaches a density of 200 kg m$^{-3}$. The evolution of the air gap thickness shows a linear progression from 0.1 mm to 3.6 mm (subplot in Fig. 9). By a visual inspection, we noted that this air gap developed at the whole base of the snow slab, and not only in the cylinders. Finally, SSA and correlation length
show a slight slope in their vertical profile, that seems to develop over time (Fig. 9). From top to bottom, SSA tends to decrease and correlation length tends to increase, so that grains become slightly larger towards the bottom.





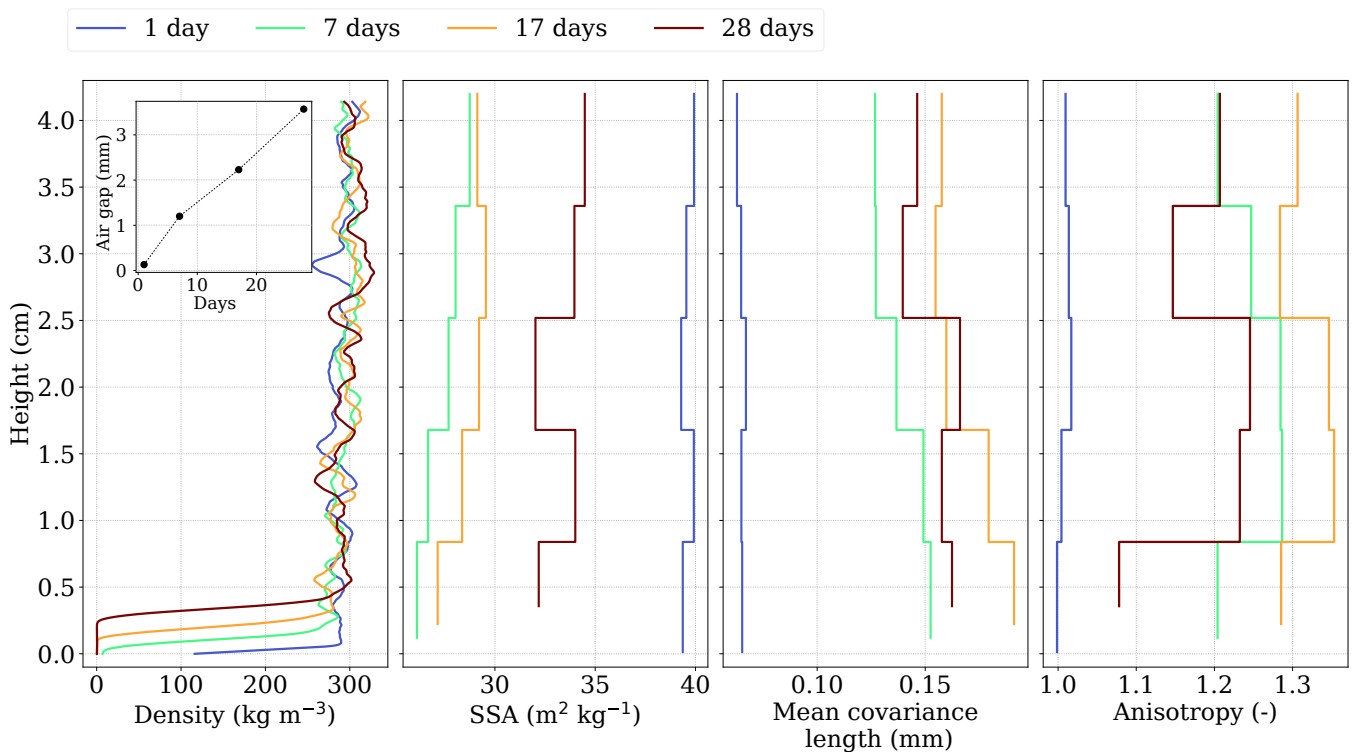

**Figure 9.** Evolution of the vertical profile of the microstructural properties over time during Experiment B. The evolution of the air gap thickness is displayed in the subplot.

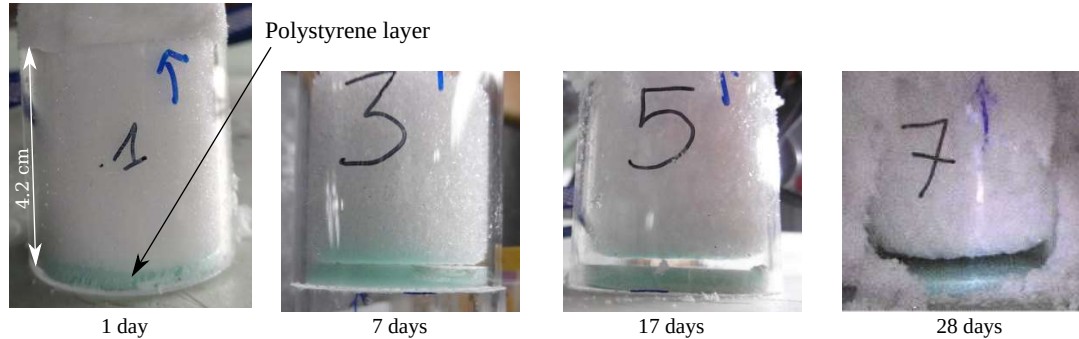

**Figure 10.** Pictures of the snow samples collected at different times from the snow layer during Experiment B. The development of the air gap can be seen at the sample base.





**Figure 11.** (a) Growth rate and horizontal slices at different heights of the sample at 20 days for Experiment A. (b) Growth rate and horizontal slices for Experiment B. (c) Nakaya diagram (adapted from Libbrecht, 2021) with colors representing the basal (orange) and prismatic (blue) regimes. On the temperature axis, the red line corresponds to the temperature range of Experiment A and the green line to the one of Experiment B (the dotted line represents the snow out of the scanned sample). The red path is a rough estimation of the supersaturation and temperature evolution in the height of Experiment A, the arrows point from the bottom (left) to the top (right).



## 3 Discussion

### 3.1 Vertical evolution of the grain morphology

In Experiment A, different grain morphologies develop over time along the vertical, with coarser grains and pores in the middle characterized by higher covariance length and SSA (Fig. 5 and 6). This distribution was not present at the initial stage and formed during the TGM. To further quantify the evolution of the ice and air heterogeneities size in the vertical direction, a growth rate was calculated at different heights by dividing the difference between the initial and the final covariance lengths by the duration of the experiment (Fig. 11.a). This $l_c$-based growth rate depicts the growth of the air and ice structure at the

REV scale. As expected, the growth rate also shows a non-monotonous distribution with a maximum in the middle part at $9.4 \times 10^{-11}$ m s$^{-1}$. As a comparison, during the experiment of Kamata and Sato (2007), the growth rate increases linearly from 0 m s$^{-1}$ at the top of the snow layer at -70°C to $8 \times 10^{-10}$ m s$^{-1}$ at the bottom at -10°C.

The more intense growth found in the center part is surprising as it does not follow the well-known dependency between metamorphism rate and temperature: the warmer the temperature, the more active the metamorphism (e.g., Kamata and Sato,

2007). In our case, largest grains would have been expected at the warmer bottom and smallest grains at the colder top, with a monotonous grain size distribution across the snow layer. Thus, temperature alone can not explained our results. Other known impacting factors are the temperature gradient intensity and snow density. As seen in Fig. 3 and 5, both parameters show a monotonous vertical distribution and can not be the cause of the observed vertical heterogeneities.

**Temperature range effects:** To explain our vertical signal, one hypothesis is the impact of the different regimes of crystal growth depending on temperature. Ice crystals grow preferentially depending on temperature either along the c-axis (basal regime) or along their a-axis (prismatic regime). Those preferential growth regimes are illustrated in the Nakaya diagram in the case of growing ice crystals in the atmosphere (Fig. 11.c), where either column-like crystals or plate-like crystals are obtained. Experiment A covers temperatures from -3.1°C to -15.6°C (in red in Fig. 11) and comes across different regimes of the Nakaya

diagram: from basal (in orange), to prismatic (in blue), to again basal, from the bottom to top of our snow layer. Interestingly, when overlapping the regimes and the growth rate profile, the variations in the profile seem to match with the three regimes (Fig. 11.a). The highest growth rates coincide with the basal regime, whereas smaller values are found rather in the prismatic regime. In terms of microstructural features, different morphologies are observed in the prismatic regimes and the basal regime. Former studies already pointed out a link between crystal growth regime and the shape of snow grains in the snowpack. Akitaya

(1974) suggested that crystal regimes influence the shape of depth hoar. Based on micro-photographs, the author observed a dependence of temperature on the depth hoar habits, such as needle-like grains at -5.1°C and plate-like grains at -2.2°C, which is in agreement with the Nakaya diagram. In terms of field observations, Sturm and Benson (1997) investigated the vertical sequence of snow texture in subarctic snow cover using a microscope and photographs. In the lower part of the snowpack, they report two layers that show opposite characteristic with rather plate shapes or rather columnar shapes, which could be

attributed to prismatic and basal growth, respectively.

To explain how basal growth could have caused the observed enhanced grain growth in the center of the layer, we suggest a sce-





nario involving the crystal orientation of the ice grains. During the sieving phase, the snow was composed of recent snow (DF) showing many plate-like crystals. Riche et al. (2013) suggested that such crystals tend to deposit with their c-axis preferentially vertically oriented because of their shapes. In our case, this would lead to numerous grains having their c-axis oriented close
to the vertical axis. A higher proportion of basal surfaces would thus face the vertical vapor flux compared to the prismatic faces. Hence, in the center of the layer where basal growth is favored, those basal surfaces would grow actively as they benefit from both a high vapor intake and enhanced phase change. In comparison, in the upper and lower part where prismatic growth is favored, prismatic surfaces do benefit from enhanced phase change but not from a high vapor intake, as they do not face the vapor flux. It would result in the formation of larger columnar grains in the center part of the snow layer in comparison to
the top and bottom. A similar orientation selective metamorphism has been observed by Granger et al. (2021) between -4 and -2°C for prismatic faces: based on diffraction contrast tomography and tracking interface, they show that prismatic preferential growth occurs, with higher sublimation and deposition rates, for grains with their c-axis close to the horizontal plane. This observation was actually realized from a nearly isotropic initial distribution of c-axis. In fact, without considering the initial orientation of the c-axis in our experiment, the temperature range effect would probably also lead to a selection of vertical
c-axes between -4 and -10°C and to horizontal c-axes elsewhere, thus providing the observed morphologies.
Unlike Experiment A, Experiment B shows a growth rate and a covariance length profile that do not show any clear bump and are rather monotonous (Fig. 9 and 11.b). In Experiment B, the temperature range varies from -14.5 to -6.5°C in the snow layer, and from -11 to -6.5°C inside the samples, meaning that most of the scanned snow underwent temperature conditions that fall within the sole basal regime. In terms of magnitude, the growth rate is lower in Experiment B than in Experiment A. This can
be related to the density, higher in Experiment B than in Experiment A, as grain growth is more constrained in space in dense snow so that smaller grains form (e.g., Akitaya, 1974; Granger et al., 2021).

**Supersaturation effects:** We discuss above the influence of the crystal growth regimes on the microstructure. The Nakaya diagram shows that, in addition to the temperature, the vapor supersaturation also has an impact on the shape of the grains
(Fig. 11.c). In conditions of high supersaturation, grains show more complex shapes such as dendritic and hollow structures. As seen in our measurements in Fig. 3, the top and bottom areas of the snow layer show higher supersaturation conditions, indicating more active sublimation and condensation processes, whereas the center area is close to the saturation, indicating a less active zone. Concordances can be found between supersaturation and our observed grain shape distribution. The red path in Fig. 11.c enables to link the Nakaya diagram, the temperature and supersaturation conditions across the snow layer, and
the observed microstructural features. From bottom to top of the layer, the final snow microstructure of Experiment A goes through small and complex dendritic grains (low covariance length and high SSA), to large and less complex structures (high covariance length and low SSA), to small grains with an intermediate complexity (low covariance length and average SSA). This is in agreement with the Nakaya diagram, which predicts, following the same path, transitions from very dendritic plates, to solid columns, and then to thin plates.
The physical processes resulting in this type of evolution could be the following: in the less active area, the low supersaturation condition could favor a slow growth of massive grains and a decrease of the number of grains. In the more active areas (top



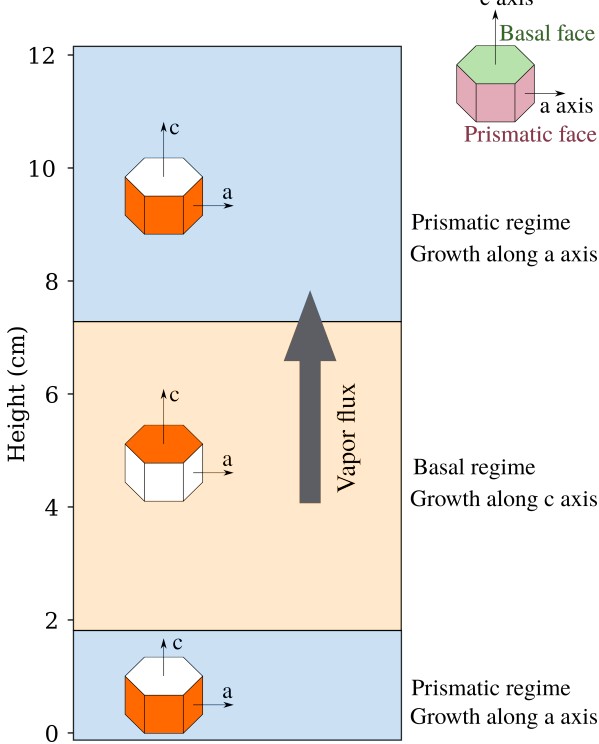

**Figure 12.** Simplified representation of grain growth configuration in the case of vertical c-axis for Experiment A.

and bottom), fast sublimation and condensation rates could come along with a rapid growth of small and complex features (dendrites), while keeping the pore and grain sizes rather small.

It is worth noting that the opposed profiles of the $l_c$-based growth rate and the supersaturation are not in contradiction as the growth rate describes the structures growth at the REV scale, thus at a larger scale than the supersaturation that generates features at much lower scale, which are not captured by our growth rate.

**Conditions for morphological heterogeneities and impacts:** In the light of both experiments A and B, it seems that specific conditions must be fulfilled to observe the formation of heterogeneities within an initial homogeneous snow layer under TGM that would be caused by the crystal growth regimes and the supersaturation variations. First of all, the temperature range should cover multiple growth regimes, which is easier to maintain in laboratory than in natural conditions, where the upper temperature is constrained by the changing atmospheric temperature. The initial density appears to play a role: low-density snow enables conditions closer to the one encountered in the atmosphere as crystal growth is not restricted by the proximity of other grains, whereas high-density snow constrains the condensation process on a shorter scale (Granger et al., 2021). The processes described here happen within a few centimeters inside the snowpack and impact greatly the microstructural properties such as the covariance length and the SSA, while keeping the density nearly constant. From an initial homogeneous



snow layer, the temperature and vapor density conditions lead to considerable heterogeneities in the layer microstructure. To describe a natural snowpack undergoing temperature gradient conditions, considering those processes could have a significant impact on macro-scale models such as Crocus (Vionnet et al., 2012) and Snowpack (Lehning et al., 2002). For instance, the
thermal conductivity is at first order driven by the density, but Calonne et al. (2011) has shown that it is also dependent on the microstructural arrangement. Similarly, the mechanical behavior of the snowpack could also be impacted by heterogeneities in the grain morphologies of a snow layer (see e.g., Hagenmuller et al., 2014; Wautier et al., 2015).

## 3.2   Basal mass loss

An important part of this study is the monitoring of the mass loss in the first millimeters of the snow layer, made possible by the non-disturbed sampling method of Experiment B. The standard sampling method, as used in Experiment A, does not provide reliable data on the bottom area of the snow (see Sec. 1.1.1). Moreover, the removal of the top copper plate disturbed the temperature field. In contrast, in Experiment B, the copper plates remained in place and the contact with the snow was insured during the whole experiment, as the samplers were extracted from the snow layer from the side of the temperature
gradient box.

The air gap on the base of the sample reached 3.6 mm, which corresponds to a mass loss of 5% of the initial ice, in 28 days for a temperature gradient about 100 K m$^{-1}$ and temperatures near -6.5°C. As only the first 4 cm of the snow layer (7 cm total height) was sampled and scanned, the mass redistribution in the upper part of the snow layer cannot be observed. This air gap was caused by the continuous mass transfer from the warm bottom to the cold top through the processes of diffusion
and phase changes. For comparison, in the extremely-high temperature gradient experiment of Kamata and Sato (2007), mass transfer was studied for a snow layer of 10 cm by weighing the mass of sub-layers of 2.5 cm height each. The density of the lower sub-layer decreases from 166 to 152 kg m$^{-3}$ in 5 days under a local gradient of 270 K m$^{-1}$ and an averaged temperature of -16°C, whereas the other three sub-layers show a slight increase of of 2 to 4 kg m$^{-3}$ under gradients ranging from 480 to 700 K m$^{-1}$ and averaged temperatures from -26 to -56°C. As the vertical resolution was 2.5 cm, it is difficult to know whether
a localized air gap formed, like in our experiment. Wiese (2017) monitored by in-vivo tomography the evolution of a snow sample laying on a artificial frozen soil (frozen glass beads) under a 100 K m$^{-1}$ gradient. After 21 days, they observed the formation of a low-density layer of 2 to 3 mm, as well as the partial drying of the soil. In terms of field observations, Domine et al. (2019) reported a gradual de-densification of the first 10 cm at the base of the Arctic snow cover from 400 to 160 kg m$^{-3}$ for snow that underwent about 4 months of temperature gradient between 30 and 50 K m$^{-1}$. In comparison to the gradual
decrease they observed, our results show a very abrupt mass loss with a thin layer completely empty of ice (maintained by the upper plate and the sides of the gradient box and the sampling tubes), which may be due to the dry condition imposed at the bottom ensuring no source of water vapor from underneath.

Depending on the soil nature and humidity, an abrupt or more gradual less-dense layer can be formed. Wiese (2017) showed that even for soils containing a large proportion of ice, a low-density layer can appear. A cavity of several centimeters on the
whole base of the snow cover could hold in spite of gravity with the topography or the vegetation making bridges on the snow





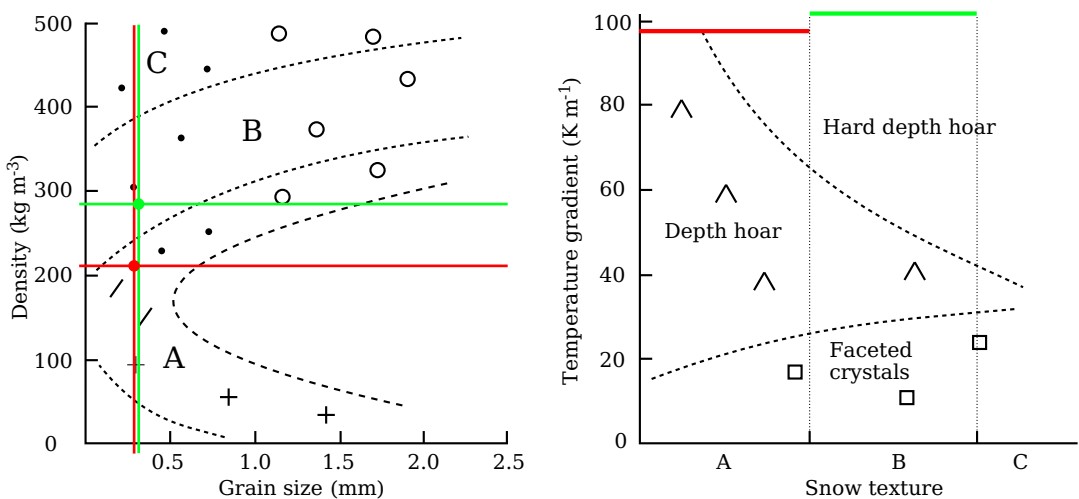

**Figure 13.** Application of Fig. 25 of Akitaya (1974) on the conditions of Experiments A (red) and B (green). This figure was used by the author to link characteristics of the initial snow to the resulting depth hoar types.

base. As suggested by Domine et al. (2019), this drop in density at the ground interface could have a major incidence on macro-scale snow models such as Crocus and Snowpack in terms of mechanical processes and heat and mass transfer properties. In this regards, the provided data, that include quantification of the mass loss as well as the well-constrained forcing conditions, can contribute to evaluate mass transfer models.

## 415  3.3  Hard depth hoar observation

When comparing the microstructure obtained at the end of experiments A and B, very different features can be observed. In Experiment A (Fig. 4), the final microstructure is standard (skeleton-type) depth hoar, which is known to be fragile. In Experiment B (Fig. 7 and 8), the final microstructure corresponds to hard (or cohesive) depth hoar, as confirmed by the hand hardness test. Another difference between experiments A and B is the trend in SSA. In Experiment B (Fig. 9), the SSA
decreases in the first week, and then increases for the following three weeks, whereas in Experiment A (Fig. 5) the SSA decreases during the whole period. The decreasing trend of SSA is an evidence of hard depth hoar development, as the rapid sublimation and deposition rate on a dense structure results in very dendritic and intricate structures that tend to increase the SSA with more complex shapes, opposed to the typical logarithmic decrease of SSA reported in most of the metamorphism studies (e.g., Cabanes et al., 2003; Flin et al., 2003; Domine, 2009). The trend in the SSA for the development of hard depth
hoar is thus controlled by two opposite mechanisms, grain growth and the formation of complex surfaces (Wang and Baker, 2014).

In terms of morphology, Akitaya (1974) reported that the three main factors determining if the TGM will produce hard depth hoar are the initial density, the initial snow type (and grain size), and the temperature gradient (Fig. 13). Following this idea, the initial conditions of Experiment A (DF, grain size $\simeq 0.3$ mm, $\rho_s = 210$ kg m$^{-3}$, TG = 93 K m$^{-1}$) could likely lead to





standard depth hoar whereas the initial conditions of Experiment B (RG, grain size $\simeq 0.3$ mm, $\rho_s = 287$ kg m$^{-3}$, TG = 103 K m$^{-1}$) would lead to hard depth hoar.

Following Wang and Baker (2014) who studied the formation of hard depth hoar for low-density snow, we showed that hard depth hoar can form not only with extreme temperature gradient or very dense snow, but also for more standard conditions with a rather strong gradient of 100 K m$^{-1}$ and a density of 287 kg m$^{-3}$. Hard depth hoar presents specific mechanical properties

with a cohesive structure compared to the typical depth hoar, making its prediction and identification valuable, for instance for avalanche risk forecasting.

## 4  Conclusion

This study focuses on two features of strong temperature gradient metamorphism at the scale of a snow layer: heterogeneous grain growth and vertical mass transfer. For that, two cold-room experiments were performed to monitor the evolution of a

snow layer held under a 100 K m$^{-1}$ temperature gradient, based on regular snow sampling and imaged by X-ray tomography. Temperature and humidity profiles were recorded during the first experiment to monitor the environmental conditions of the snow layer. We report that, from an initially homogeneous snow, strong heterogeneities have developed on the vertical dimension of the snow layer. Coarser and more solid grains developed in the middle of the layer compared to the top and bottom part, as showed by a covariance length about three times higher and a lower SSA. Transitions in microstructure seem to match

with transitions in crystal growth regime (Nakaya diagram). Indeed, snow is subjected to temperatures that favor, from bottom to top, prismatic, basal, and prismatic growth regimes. The temperature dependency on the Nakaya diagram is also interlinked with the supersaturation axis of the diagram. Our supersaturation measurements depict an area close to saturation in the middle and supersaturated in the top and bottom part. These conditions could lead to coarse grains in the middle and to small complex ice features elsewhere, as observed in our experiment. The other significant vertical signal was the basal mass loss, observed

thanks to a non-disturbing sampling method. An air gap of more than 3 mm formed in 28 days. This drastic mass loss was caused by the vertical vapor flux and accentuated by the dry boundary conditions imposed at the bottom. Finally, from two similar experiments, hard depth hoar was observed in the second experiment characterized by higher initial density. Hard depth hoar was identified through a hand hardness classified as "pencil" and shows an increasing SSA with time, opposed to the typical trend.

Our results show that temperature gradient can form heterogeneities from an initial homogeneous layer, regarding mass and morphology. As a consequence, macro-scale processes would also be impacted, such as heat transport or mechanics. Hence, it could be important to include these temperature gradient features in the snowpack models to refine the description of the metamorphism, transport properties, and mechanical behavior. To go in this direction, our datasets, composed of tomographic time series associated with temperature and humidity profiles, constitute a valuable evaluation dataset with well-constrained

conditions and quantified evolution. Especially, temperature, humidity and mass distributions can be used to evaluate heat and mass transfer models. The supersaturation profile constitutes so far the first experimental estimation of the supersaturation from humidity sensors inside a snow cover. It queries the approximation of the water vapor density equal to the saturation



water vapor density in a snow cover, that does not seem suited for strong temperature gradient conditions, especially in the top and bottom areas. To further quantify the basal mass loss, a systematic study could be done, considering the intensity of

the temperature gradient and the soil moisture influences. Complementary studies should be done to test the hypothesis provided for the heterogeneous depth hoar growth, adding crystalline orientation measurements for instance. Another interesting possibility could be to find suitable field conditions to look for evidence of those heterogeneities, such as the high temperature gradients of the arctic snowpack.

*Data availability.*  The experimental datasets are available upon request by contacting the corresponding author.

*Author contributions.*  LB and NC designed the experiments and carried them out. The data were analyzed and interpreted by LB with the help of FF, NC and CG. LB prepared the manuscript with contributions from all co-authors.

*Competing interests.*  The contact author has declared that none of the authors has any competing interests.

*Acknowledgements.*  The authors would like to thank L. Pezard and J. Roulle for their help during the experiments. The tomography apparatus (TomoCold) was funded by INSU-LEFE, Labex OSUG (Investissements d'avenir – ANR-10-LABX-0056) and the CNRM. The 3SR lab is

part of the Labex Tec 21 (Investissements d'Avenir, Grant Agreement ANR-11-LABX-0030). CNRM/CEN is part of Labex OSUG@2020 (Investissements d'Avenir, Grant ANR-10-LABX-0056). This research has been supported by the MiMESis-3D ANR project (ANR-19-CE01-0009).



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



**Table 2.** List of the 30 tomographic images used in experiments A (above) and B (below). Snow types are given according to the international classification (Fierz et al., 2009).

| Sample name | TG duration (days) | Size: x × y × z (voxels) | Voxel size (μm) | Snow type | Density (kg m$^{-3}$) |
|---|---|---|---|---|---|
| t0_big | 0 | 1000 × 1000 × 6200 | 21.7 | DF | 184.1 |
| t0_small_top | 0 | 1200 × 1200 × 1200 | 8 | DF | 204.6 |
| t1_big | 1 | 1000 × 1000 × 5700 | 21.7 | RG/FC | 189.7 |
| t1_small_top | 1 | 1200 × 1200 × 1200 | 8 | RG/FC | 200.1 |
| t1_small_bot | 1 | 1200 × 1200 × 1200 | 8 | RG/FC | 228.3 |
| t2_big | 2 | 1000 × 1000 × 5700 | 21.7 | FC | 208.9 |
| t2_small_top | 2 | 1200 × 1200 × 1200 | 8 | FC | 202.9 |
| t2_small_bot | 2 | 1200 × 1200 × 1200 | 8 | FC | 219.8 |
| t3_big | 5 | 1000 × 1000 × 5600 | 21.7 | FC/DH | 213.6 |
| t3_small_top | 5 | 1200 × 1200 × 1200 | 8 | FC/DH | 213.6 |
| t3_small_bot | 5 | 1200 × 1200 × 1200 | 8 | FC/DH | 227.3 |
| t4_big | 7 | 1000 × 1000 × 5550 | 21.7 | DH | 227.4 |
| t4_small_top | 7 | 1200 × 1200 × 1200 | 8 | DH | 224.6 |
| t4_small_bot | 7 | 1200 × 1200 × 1200 | 8 | DH | 238.4 |
| t5_big | 9 | 1000 × 1000 × 5550 | 21.7 | DH | 219.7 |
| t5_small_top | 9 | 1200 × 1200 × 1200 | 8 | DH | 213.0 |
| t5_small_bot | 9 | 1200 × 1200 × 1200 | 8 | DH | 233.5 |
| t6_big | 12 | 1000 × 1000 × 5700 | 21.7 | DH | 218.8 |
| t6_small_top | 12 | 1200 × 1200 × 1200 | 8 | DH | 235.2 |
| t6_small_bot | 12 | 1200 × 1200 × 1200 | 8 | DH | 241.7 |
| t7_big | 14 | 1000 × 1000 × 5450 | 21.7 | DH | 224.8 |
| t7_small_top | 14 | 1200 × 1200 × 1200 | 8 | DH | 241.9 |
| t7_small_bot | 14 | 1200 × 1200 × 1200 | 8 | DH | 226.9 |
| t8_big | 20 | 1000 × 1000 × 5500 | 21.7 | DH | 228.6 |
| t8_small_top | 20 | 1200 × 1200 × 1200 | 8 | DH | 227.1 |
| t8_small_bot | 20 | 1200 × 1200 × 1200 | 8 | DH | 223.3 |
| t0 | 1 | 1300 × 1300 × 4200 | 10 | RG | 288.3 |
| t1 | 7 | 1300 × 1300 × 4200 | 10 | DH | 291.5 |
| t2 | 17 | 1300 × 1300 × 4200 | 10 | DH | 293.5 |
| t3 | 28 | 1300 × 1300 × 4200 | 10 | DH | 296.2 |