# Peer review of "Heterogeneous grain growth and vertical mass transfer within a snow layer under temperature gradient"

_The Cryosphere, 2022_

## Referee Comment (RC1)

General comments

This paper conducted cold chamber experiments for continuous observation of snow metamorphism under the strong temperature gradient approximately 100 K/m and clarified the heterogeneous grain growth. As the authors say, there have been several previous studies in which snow layer was continuously observed in a low-temperature chamber using X-ray CT, but the novelty of this paper lies in the experiment that the vertical structure of the approximately 10 cm snow layer was observed precisely and continuously. By applying a strong temperature gradient to a thick snow layer, the different regimes of crystal growth within a single snow layer were achieved. The authors also mentioned a decrease in the density at the bottom of the snow layer, and the formation of hard depth hoar. These results will provide fundamental data for snow transformation modeling and snow stability prediction.

Specific comments

1. The authors mention that different regimes of crystal growth were observed depending on the temperature range. Figure 11 shows in the height direction for each temperature zone, but the columns and plates are very difficult to see from the 2D image. Photographs of the particles or a 3D surface rendering would be appreciated.

2. Sturm and Johnson (1991) reported that the depth hoar near the bottom of the natural snowpack in Alaska has a C-axis that is oriented almost horizontally in some places and is growing horizontally. (same figure of Fig. 2 of Sturm and Benson (1997)).
In this range of prismatic growth, did the prismatic face of the snow with the C-axis oriented horizontally grow vertically and the horizontal basal growth was not observed?

3. The experiment with large temperature gradients at very low temperatures is similar to that of Kamata and Sato (2007). However, Kamata and Sato's experiment lasted 5.5 days, whereas the authors observed for about a month. The authors would mention what differences they found over a longer period of time, although there is a description of a large change in the initial period.

4. The differences in temperature ranges appearing in the long vertical samples lead to interesting results. However, since the number of experiments was only two, it is hoped to increase the number of experiments to obtain a data set in future work. Kamata and Sato also have a few experiments, so these experiments will provide valuable data.

Technical corrections
Line 77
"Those evolution" is "Those evolutions"

Line 306
"can not explained" is "can not explain"

Line 590
"Yosida: " is "Yosida, Z."

---

## Author Comment (AC1)

**Authors' reply to referee comments RC1**

We thank very much Reviewer 1 for the comments that help improving the manuscript. Please find below our point-by-point replies in green color.

General comments

This paper conducted cold chamber experiments for continuous observation of snow metamorphism under the strong temperature gradient approximately 100 K/m and clarified the heterogeneous grain growth. As the authors say, there have been several previous studies in which snow layer was continuously observed in a low-temperature chamber using X-ray CT, but the novelty of this paper lies in the experiment that the vertical structure of the approximately 10 cm snow layer was observed precisely and continuously. By applying a strong temperature gradient to a thick snow layer, the different regimes of crystal growth within a single snow layer were achieved. The authors also mentioned a decrease in the density at the bottom of the snow layer, and the formation of hard depth hoar. These results will provide fundamental data for snow transformation modeling and snow stability prediction.

Specific comments

1. The authors mention that different regimes of crystal growth were observed depending on the temperature range. Figure 11 shows in the height direction for each temperature zone, but the columns and plates are very difficult to see from the 2D image. Photographs of the particles or a 3D surface rendering would be appreciated.

In experiment A, the covariance length in the middle of the snow layer increased faster than in the top and bottom. We hypothesized that this differences in grain growth could be due to the effect of the crystal growth regimes, temperatures in the middle part falling in the basal growth regime whereas temperatures in the top and bottom parts falling in the prismatic growth regime. Hence, the impact of different regimes of crystal growth was identified through differences of growth rate, and of the overall microstructure, but not based on differences in crystal habits, i.e. individual column-like or plate-like crystals. Indeed, as mentioned by the reviewer, the snow structure is really intricated and no isolated columns or plates were observed in the snow layer. This can be seen in the 2D images (Figure 4) and 3D images (Figure 6). This contrasts with the crystals reported in the Nakaya diagram and in Akitaya et al . 1974, which present clear crystal habits. However, such crystals were observed in air or large pore space. We believe that within snow, grain growth is constrained in space by the surrounding grains, which may prevent the growth of fully-developed columns or plates. Yet, it seems that in Arctic snowpack, snow with clear crystal habits can be observed, as presented in Sturm and Johnson, 1991. Dedicated work should be done to further investigated this topic, for example lower snow density, longer and more severe temperature gradient could be reproduced to closer mimic Arctic conditions.

2. Sturm and Johnson (1991) reported that the depth hoar near the bottom of the natural snowpack in Alaska has a C-axis that is oriented almost horizontally in some places and is growing horizontally. (same figure of Fig. 2 of Sturm and Benson (1997)).

In this range of prismatic growth, did the prismatic face of the snow with the C-axis oriented horizontally grow vertically and the horizontal basal growth was not observed?

We agree with the Reviewer that, in the prismatic growth areas of the snow layer (top and bottom) one could expect an enhanced growth of the prismatic faces in the vertical direction, which is the direction of the temperature gradient. In our study, it is however not possible to observe such detailed grain evolution, as we would need to access the grain crystalline orientations as well as in vivo monitoring to track the same grains over time, as the analysis by Granger et al. 2021 with diffraction contrast tomography. Here we can only comment that indeed snow structures grew preferentially in the vertical direction compared to the horizontal one (increase of the anisotropy coefficient), but without being able to distinguish whether this vertical growth was rather occurring on prismatic or basal faces. As mentioned in the previous comment, we did not observed sufficiently representative column or plate crystals from which crystalline orientations could have been guessed, as in Sturm and Johnson 1991.

3. The experiment with large temperature gradients at very low temperatures is similar to that of Kamata and Sato (2007). However, Kamata and Sato's experiment lasted 5.5 days, whereas the authors observed for about a month. The authors would mention what differences they found over a longer period of time, although there is a description of a large change in the initial period.

Besides the experiment duration, others parameters differ between Kamata et al. 2007 and our work. In Kamata et al. 2007, the temperature gradient is much higher (530 versus 100K/m), which led to faster metamorphism. On the other hand, the temperatures imposed by Kamata et al. 2007 are cooler than ours (from -12 to -65°C versus our experiments (from ~ -3 to -17°C), which would tend to inhibit metamorphism. Thus it is not straightforward to compare our studies in terms of final metamorphism stage, taking into account temperature gradient, mean temperature and duration.
In our work, the month-duration of our experiment allowed to obtain a clear signal in the microstructure evolution: the snow layer differentiation for Experiment A and the specific surface area increase resulting from the hard depth hoar formation for Experiment B, as described in result section of the paper. Both features were hardly observable in the first to intermediate stages of the evolution.

4. The differences in temperature ranges appearing in the long vertical samples lead to interesting results. However, since the number of experiments was only two, it is hoped to increase the number of experiments to obtain a data set in future work. Kamata and Sato also have a few experiments, so these experiments will provide valuable data.

The work of Kamata et al. 1999 was included in the introduction to mention the temperature and temperature gradient effect on depth hoar formation. We agree that additional experiments would be needed to further investigate the impact of temperature and crystal growth regime on snow within a snow layer, as mentioned in the conclusion of the paper.

Technical corrections

Line 77

"Those evolution" is "Those evolutions"

Line 306

"can not explained" is "can not explain"

Line 590

"Yosida: " is "Yosida, Z."

All the above technical comments were taken into account and modified as suggested in the new version of the manuscript.

---

## Author Comment (AC2)

**Authors' reply to referee comments RC2**

We thank very much Reviewer 2 for the comments that help improving the manuscript. Please find below our point-by-point replies in green color.

**General comments**

The paper by Bouvet et al. discusses two laboratory experiments where snow is put under a temperature gradient typical for arctic conditions. The changes in snow structure are mainly analysed by micro-tomography. Similar experiments have been conducted since 2004 by other authors. Using humidity sensors, an attempt was made to measure relative humidity inside the samples. The presentation of the data is sufficient. However, the data can not be publicly assessed, and the deposition of the data in a repository is now a scientific standard.

In the revised version of the manuscript, we will provide the data obtained from our work as .ods files in the supplement.

As observed by others, snow sublimates at the warm side of temperature gradient experiments, forming an air gap (already observed by Nakaya in the 1950'ies). Such an air gap immediately caused the thermal conductivity to be reduced to the one of air, and the initially vertical and parallel heat flux became distorted as the samples were surrounded by higher conducting plastic. As much as the reviewer could see, this fact was not taken into account (e.g. by numerical simulations) for the interpretation of the structural evolution of the snowpack.

The temperature field of the snow layer in Experiment B was simulated using the software COMSOL to evaluate the effects of the plastic sample holder and the polystyrene layer. The simulation of the initial stage of the experiment is presented for two setups, with and without the underlying polystyrene plate (Figure 2.c in the paper). The simulations of the temperature field at the end of the experiment with an air gap of 3 mm height is shown in the figure below. We see that the temperature field is slightly more perturbed by the plastic holder with the air gap than without. Without air gap, the horizontal gradient represents 10% compared to the vertical one, with air gap, this ratio rises to 12.6%, which we still consider small. In addition, we stress that the effect of the plastic holder on the temperature is observed right next to the side of the holder and vanishes while moving towards the center; tomography was performed on the snow volume located in the center of the holder only.
As suggested by Reviewer 2, the new version of the manuscript includes a comment on the simulations of the temperature field with the air gap in Section 2.2. It now includes "In addition, Figure 2.c shows that the presence of the plastic cylinder does not disturb significantly the temperature field and that non-vertical gradients represent 10% compared to the vertical one. Similar results are observed when simulating the end of the experiment with an air gap (12.6%)."

[Figure]

*Figure 1. Numerical simulations of the temperature field for the snow layer at the final stage of Experiment B considering an air gap of 3 mm height.*

The interpretation of the temperature and humidity profiles is consequently misleading. Without considering the non-vertical heat fluxes, no valid conclusions are possible. The sections "Results" and "Conclusion" must be rewritten and re-interpreted, considering a heterogenous temperature gradient and heat flux.

This comment might be the result of a misunderstanding of our experiments. We present two experiments, Exp. A and Exp. B, that differ in their set-up. Temperature and humidity were measured only in Exp A and not Exp B. The sampling method, to collect snow for tomography, was different in both experiments. Plastic cylinders were initially buried in the snow layer for further sampling in Exp B but not in Exp A. The latter followed the standard snow sampling procedure. Hence, the temperature and humidity profiles in Exp A are not affected by the presence of plastic cylinders as there was none. Concerning Exp B, as explained in the comment above, non-vertical heat fluxes related to the plastic holder were insignificant compared to the vertical ones and concentrated right next to the holder's sides. Besides, Exp B focused on the analysis of the air gap formation that was consistently observed everywhere at the bottom of the layer, inside and outside of the sample holders.

**Specific comments**

No details and data are given on how the humidity sensors are calibrated at below zero-degree conditions. Calibration before and after the experiment would have been necessary to have valid data.

Additional information on the calibration of the humidity sensors were included in the new version of the manuscript, which reads: "The SHT25 sensors are marketed with a humidity calibration at ambient conditions (~ 20°C and ~ 50% RH) and present large offsets when placed in cold and humid conditions, up to 7% RH. A calibration was conducted by placing the sensors in snow to reach close to vapor saturation

conditions, in a temperature controlled box. The applied conditions varied between -4°C and -14°C, and between 85% and 100% RH. A HMP110 (Vaisala) sensor was used as reference humidity value. As the humidity error is correlated to the temperature, a linear correction was applied. In our range of temperature, the correction was between 0 and 8%."

Also, having done further analysis on the sensors data, we decided to remove the SHT15 data because of acquisition errors, and to only use the PT100 data for the temperature, and the SHT25 data for the humidity (as the SHT25 temperature sensors are less accurate). Because of the humidity uncertainty (increased by the removal of one set of sensors), we decided to shorten the analysis of the vapor supersaturation in the new version of the manuscript.

The authors state that the initial density is almost constant. Their figs. 5 and 9 clearly show density fluctuations of up to about 30% at a distance of a few millimetres. Such density variations strongly affect thermal gradients and, therefore, snow metamorphism. A detailed interpretation of thermal conductivity and temperature gradients is necessary.

Figure 5 and 9 of the paper show the vertical profile of snow density computed on a 2.1 mm moving windows, for Experiment A and B, respectively.  At this resolution, density variations up to 30% can be seen for the Experiment A, mostly at the initial stage as they smooth out with time. Those variations were formed during the sieving process to create the snow layer. Sieving induces vertical variations as well as spatial variations within the snow layer. Those initial density variations at the mm-scale have no major impact on snow metamorphism as no related variation at the mm-scale was found in the specific surface area or the correlation length, which both reflect metamorphism. In the paper, the significant result that is pointed out is the development of coarser grains (higher correlation length, lower specific surface area) in an area of ~ 4 cm height located in the middle of the snow layer. This temporal evolution on a cm-scale area of the snow layer shows no correlation with the density fluctuations, observed at a mm-scale. The effect of mm-scale density fluctuations was thus of second order for the metamorphism described here.

As pointed out by Reviewer 2, we clarified the description of the vertical profile of density to include the mm-scale fluctuations, such as "The overall density remains constant along the vertical, around 220 kg m$^{-3}$, although significant initial vertical variability can be observed, mainly caused by the sieving process."

The mean covariance length (which should probably read "mean correlation length") is given without a directional index, and no formula or precise reference is given for its calculation. Is the mean covariance length averaged in the horizontal and vertical directions?

Yes, the mean covariance length is the averaged of the covariance length in the x-, y-, andz- direction. We improved the description of the mean covariance length in the new version of the manuscript, such as "The covariance (or correlation) length $l_c$, which corresponds to the characteristic size of a heterogeneity made of an ice grain and a pore, was calculated along the x-, y- and z- directions of the images, as in the work of Calonne et al. 2014. In this study, we mainly use the average of $l_c^x$, $l_c^y$, and $l_c^z$, referred as the mean covariance length in the following."